# Feasibility of Using Facebook to Engage SNAP-Ed Eligible Parents and Provide Education on Eating Well on a Budget

**DOI:** 10.3390/ijerph19031457

**Published:** 2022-01-27

**Authors:** Kristen Lawton, Lindsey Hess, Heather McCarthy, Michele Marini, Katie McNitt, Jennifer S. Savage

**Affiliations:** Center for Childhood Obesity, The Pennsylvania State University, University Park, State College, PA 16802, USA; lbb135@psu.edu (L.H.); hmccarthy@cseop.org (H.M.); mem44@psu.edu (M.M.); kmm6054@psu.edu (K.M.); jfs195@psu.edu (J.S.S.)

**Keywords:** Facebook, social media, low-income, nutrition education, food resource management

## Abstract

This study examined the use of Facebook to provide education on food resource management and healthy eating on a budget to parents of preschool aged children participating in Head Start. A convenience sample of 25 parents participated in a Facebook group based on Sesame Street’s Food for Thought: Eating Well on a Budget curriculum over a 3-week period. Parent engagement was assessed by examining views, likes, and comments on posts. Qualitative data were used to assess knowledge, attitudes, and barriers experienced related to healthy eating on a budget. The results suggest that parents were engaged throughout the intervention, as evidenced by views, likes, and comments on Facebook posts, as well as by study retention (90%). Interactions with the intervention materials varied by post content, with discussion questions having the highest level of interaction. Facebook was found to be a feasible platform for delivering the intervention, and the Facebook-adapted version of the Sesame Street curriculum was shown to engage Head Start parents living in rural areas. Further research should explore the use of social media platforms for delivering nutrition education interventions to rural populations that are otherwise difficult to reach.

## 1. Introduction

Early nutrition predicts a number of positive health, social, and cognitive outcomes [1]; however, many low-income children under the age of five years do not meet the daily nutrition recommendations [2]. Caregivers shape their child’s food preferences, eating behaviors, and food intake by serving as “gatekeepers” to what food is brought into the home. What is purchased is influenced by economic determinants [3]. The purchase of nutrient rich foods, such as fruits and vegetables, whole grains, and lean proteins, is partially influenced by food cost, especially among lower income, economically disadvantaged households [4,5]. Together, these data suggest a critical need for nutrition education related to food budgeting, resource management, and meal preparation strategies that support low-income caregivers to choose, provide, and prepare healthy, nutrient-dense foods to their families [6]. 

Head Start is a federally funded program of the United States Department of Health and Human Services that provides comprehensive early childhood education, health, and nutrition services to low-income children and their families [7]. The program serves nearly 1 million low-income (≤100% of poverty) children and families nationwide and plays an integral role in supporting the development of healthy eating and positive behaviors for children 3–5 years. Head Start aims to equip and educate caregivers with tools to better care for their children through the provision of nutrition services and education. Despite the focus on caregiver engagement and nutrition education, many low-income caregivers face barriers toward attending in-person interventions, and participation in these efforts is low [8]. In rural areas, these barriers include limited transportation access, time constraints, tight work schedules, and personal health challenges [9,10,11].

Due to the aforementioned barriers, there is a need to explore innovative and cost-efficient methods to engage low-income caregivers. Technology is one potential solution to common barriers, due to the low cost and generally high accessibility [12]. According to social learning theory, interventions may be more effective if provided in shorter doses and when available “on-demand” to accommodate the lifestyles of low-income persons [13]. Modern communication channels (e.g., social media) allow intervention doses to be provided with greater frequency and in shorter durations, and they provide an effective way to disseminate nutrition education and engage low-income parents [14]. Facebook is the biggest social network worldwide, with 2.85 billion monthly active users [15], and it is the most frequently used social media site among 18–49 year olds [16]. Mothers, in particular, are heavily engaged in social media, both receiving and providing support to others [17]. Facebook provides a platform where users can motivate each other to reach their health goals and is a source of health information [18].

The aim of the current study was to examine the feasibility of using Facebook as a platform to (1) increase the engagement of caregivers with children enrolled in Head Start and (2) provide nutrition education related to food resource management and healthy eating on a budget. In an effort to improve dietary quality and to address common barriers experienced with reaching Head Start caregivers, we examined the feasibility of using a Facebook adaptation of Sesame Street’s online curriculum, Food for Thought: Eating Well on a Budget.

## 2. Materials and Methods

### 2.1. Participants

Participants were caregivers (herein after, parents) enrolled from a convenience sample of 363 Head Start families in 4 Head Start agencies throughout 7 rural Pennsylvania counties. Of those 363 families, 214 indicated interest in participating in future studies. Interested families were mailed letters to provide information about the study and later received a phone call and/or email to provide further information. The goal was to obtain the consent of a total of 30 parents for a single Facebook group in order to support interactions and bonding among participants in the group, based on similar studies using social media for group interventions [19]. Once 30 eligible participants were identified, recruitment stopped. Participants viewed the informed consent online and completed an online screener to determine eligibility. Eligible parents were ≥18 years of age, were the primary caregiver of a child enrolled in Head Start within the last 18 months, ate at least one meal per day with that child, regularly did the grocery shopping for their family, had reliable internet access at home, regularly used an email account, and were willing to use Facebook daily for 3 weeks during the study period. 

### 2.2. Protocol

At baseline, 30 participants completed an online survey to assess parent demographics on race/ethnicity, education, income, age, and participation in federal food assistance programs. Next, participants received an invitation to a closed Facebook group (e.g., membership is by invitation only and the group is private in that posts are only viewable by group members) by research staff. The initial post from a research assistant welcomed participants to the group and outlined group rules (e.g., prohibition of strong language/cursing, not selling goods and services, and keeping the content of posts relevant to the discussion). 

Participants were encouraged to log in to Facebook daily and interact with posts by liking, commenting, and voting in polls. The research staff monitored the group daily. Protocol was developed in advance to handle rule violations. Participants who violated rules received one warning via email from the group facilitator and, after a second violation, were notified and removed from the group. Other than monitoring for misconduct, participant discussions were not monitored and curated for content. Upon completion of the intervention, participants completed an online post-survey, which was identical to the baseline survey except for the removal of demographic questions and the addition of acceptability questions. Acceptability questions were adapted from a survey used in the previous evaluation of the Food for Thought curriculum [20]. Participants were mailed a $10 and $25 gift card for completion of the baseline and post-surveys, respectively. Surveys were developed using REDCap, and all data were housed on the REDCap secure server (REDCap v 8.1.19; Vanderbilt University Medical Center, Nashville, TN, USA) [21]. The study was approved by the Institutional Review Board of The Pennsylvania State University (Protocol code 00009331, approved on 1 February 2019). Informed consent was obtained from all subjects involved in the study.

### 2.3. Intervention

The Facebook intervention spanned 3-weeks. Buffer (2018), a social media management application, was used to prepare and schedule posts ahead of time. Intervention posts were shared 5 days a week (Monday through Friday) and 2 to 3 times per day. Intervention posts were adapted from Sesame Street’s Food for Thought: Eating Well on a Budget multimedia curriculum that was designed to support and educate parents of children between the ages of 2–8 years who may have limited access to affordable and nutritious food [22]. Food for Thought is available for free online and provides videos, reading materials, and resources (e.g., tip sheets, grocery shopping list templates, etc.) related to making healthy food choices on a budget. Prior evaluation of the curriculum revealed that participants found the Food for Thought materials to be useful, appealing, and easy to understand and also showed an impact on participants’ knowledge, attitudes, and behaviors regarding how to cope with food insecurity and how to develop and maintain healthy habits [20].

Adaptation of the curriculum to a Facebook platform was accomplished through a number of strategies. First, content was rephrased from declarative nutrition education messages to discussion questions in an effort to engage parents in conversation. Videos, recipes, and resources were provided as stand-alone posts. Finally, polls (posts that allowed participants to vote) were created to engage participants with the content. There were a total of 31 posts: 27 intervention posts and 4 non-intervention posts (e.g., welcome post and survey completion reminders). Among the intervention posts, many were interactive in nature. Seven posts included videos, three involved polls that prompted participants to vote, and nine provided links that gave participants the ability to access tip sheets, shopping lists, and handouts that they could print out at home. The remaining eight posts were informative in nature (e.g., making a shopping list, stretching your food dollars, eating well on a budget, budget-friendly cooking tips and recipes) and prompted discussion by asking questions about the content. Examples of materials provided to participants from the curriculum are shown in Figure 1. 

### 2.4. Data Analysis

Descriptive statistics were calculated for demographic variables using SAS version 9.4 (SAS Institute Inc., Cary, NC, USA). Feasibility outcomes included retention, engagement, and acceptability of the intervention. Engagement was assessed using Sociograph, a Facebook analytic tool, which summed all interactions with posts (e.g., reactions/likes, comments, and votes). Sociograph provides a sample-specific rating for each post to determine the level of engagement by using a formula that considered the number of likes, comments, and shares (Rating formula = Likes × 2 + Comments × 3 + Shares × 5). Higher ratings indicated relatively greater influence and engagement of the post. Two research staff independently logged the number of views, comments, and likes for all posts. Double-entered data were compared by a third research staff for any inconsistencies. Sustained engagement was evaluated by assessing the percentage of participants who interacted with the last post of the intervention. Self-reported participant engagement with the posts was also assessed using multiple choice questions on the post-survey about how often participants viewed group content. The retention rate was calculated as the rate of completion of the follow-up assessment. Acceptability was assessed by examining participant responses to questions on the post-survey that asked participants to rate the intervention and answer whether or not they would suggest a similar group for parents of Head Start children.

Open-ended survey responses were analyzed using a thematic analysis that involved six phases: familiarization with the data, generation of initial codes, construction of themes, reviewing themes, naming themes, and producing a final report [23]. After familiarization with the data, a researcher with graduate training in qualitative data analysis coded the data to identify potential themes. Analysis was inductive and followed the constant comparative method until themes were generated and defined [24]. The content of the three open-ended questions was developed to gain insight into anything the participant learned, what (if anything) they found most useful, and to elucidate whether they would recommend a similar group to other Head Start parents. The questions included: “What is the most useful thing that you learned from the Food for Thought Facebook group?”“Are you doing anything new with your family that you were not doing before you joined the Facebook group? Please explain.”“Would you recommend joining a similar group for other caregivers of Head Start preschoolers? Why or why not?”

## 3. Results

Thirty participants completed the baseline survey and were granted access to the Facebook group. Among these 30 participants, 25 participated in the Facebook group. A majority of participants were female (n = 25, 100%), white and Non-Hispanic (n = 21, 84%), and parents of the Head Start child (n = 22, 88%). The average age was 31 years (SD ± 4.47 years). A majority (n = 20, 80%) received SNAP (Supplemental Nutrition Assistance Program) benefits and 48% (n = 12) participated in the Special Supplemental Nutrition Program for Women, Infants, and Children (WIC). Additional demographic data can be found in Table 1.

### 3.1. Participant Engagement

Among the 25 participants in the Facebook group, 23 completed the post-survey, indicating a 92% retention rate. One participant was removed during the study period due to noncompliance with group rules. An additional participant was lost to follow-up. Engagement as assessed by Sociograph was sustained through the end of the intervention, with 22 (88%) participants interacting with the last post of the intervention. Participants interacted with every post by commenting, liking, or voting in the polls. Over the 3-week intervention period, the mean (SD) number of likes, comments, and votes per post were 9.6 (10.8), 6.6 (8.3), and 1.8 (1.2), respectively (Sociograph). About half of the participants (n = 12, 48%) reported visiting the Facebook group at least once a day, 8 (32%) reported that they visited multiple times per day, and 3 (12%) responded that they visited every other day. The total number of interactions varied by participant. A majority (44%) of participants had 20 or more interactions, 28% had 8–20 interactions, and another 28% only had 0–8 interactions with the posts. Of the 7 participants who had the fewest interactions with posts (0–8 interactions), three of these participants solely viewed the posts and did not actively engage by liking, commenting, or participating in polls. Discussion question posts were the most viewed post-type, followed by polls, videos, informative posts, and links. Videos received the most likes, followed by informative posts, links, discussion questions, and polls. Multiple engagements by all members of the group were also examined by post type, as shown in Table 2.

Rating data from Sociograph revealed that seven posts had a rating of “40” or higher, indicating that those posts had the greatest influence. Five of the seven posts were discussion questions that prompted participants to answer, one was a video, and the post with the greatest influence was initiated by a participant (i.e., “What is the age of your child?”). Sixteen participants commented on that post, defining it as the post with the greatest influence. In total, there were four posts that were initiated by participants (i.e., “Happy Mother’s Day”, “What is good for a pregnant woman to eat with twins?”, “My 7-year-old tried snap peas last night and he said he wasn’t too fond of them, but his reaction to trying something new was incredible”).

### 3.2. Acceptability & Feasibility

All participants (n = 25, 100%) answered that they would recommend a similar Facebook group to other parents with preschool children. Twenty-three participants reported that the Facebook group was easy to use, and one answered that it was a little difficult. A majority (n = 21, 91%) reported that the intervention was useful, and 17 (74%) reported that materials were easy to understand. Fifteen (65%) answered that they used the recipes from the intervention, and a majority (n = 22, 96%) watched the videos one or more times. On a scale of 1–5 with “1” indicating they loved the group and “5” meaning they disliked the group, nine (39%) answered “1”, and the mean (SD) was 1.7 (0.63). When asked how much they learned, 21 (91%) reported that they learned a lot or some things from the group, and two responded that they didn’t learn much. When asked if they would recommend joining a similar group to other caregivers of Head Start preschoolers, all 25 answered “yes”. All comments from the participants were positive.

Regarding feasibility, the implementation process went as planned, and the established protocols were successful. Six full-time research staff members monitored the group. For each day of the intervention, one staff member reviewed the page approximately every hour to monitor posts and watch for any violations of group rules. Staff members also received notifications when group members posted or commented on the Facebook page, which prompted the staff members to review the page. A back-up person was available in case of incident. The point-person checked the page regularly from approximately 7 AM until 10 PM for a total of about 1 h of staff time daily (about 21 total staff hours for the 3-week intervention). If an incident occurred, the point-person contacted the back-up staff person to review protocol and determine action steps. During this intervention, two incidents occurred, which were both flagged within 1 h of occurrence, indicating that the protocol to monitor discussions and page activity worked as intended.

### 3.3. Responses to Open-Ended Questions

Three main themes emerged from the participant responses to the open-ended questions: (1) strategies to help children try new foods, (2) meal planning and budgeting, and (3) support from peers. 

When asked about the most useful thing they learned, the most common theme was how to get their child to try new foods, especially related to picky eating. Eight parents (35%) mentioned that advice for addressing picky eating was the most useful thing they learned, with one parent saying that they learned “how to dress up foods for their picky eater to try”. When asked if they were doing anything new with their family as a result of the intervention, the top theme was trying new methods of getting their child to taste new foods. Eleven parents (48%) mentioned that they were encouraging their children to try new and healthier foods. One parent said, “Trying different healthier foods and recipes for my family. Trying to limit certain foods and instill healthier eating for my family”.

Parents expressed that the intervention taught them how to better budget and plan for meals. When asked about the most useful information they learned, eight parents (36%) mentioned that they learned more about meal planning and budgeting. One parent said, “It helped me to better budget, plan, spend more time with my kids, and got my kids eating better. I think every parent should be a part of something like this if they are not already.” When asked if they were doing anything new as a result of the intervention, five parents (22%) responded that they were taking more time to plan meals. One participant responded, “…it really does make a difference to meal plan for the upcoming week and stick to that plan when at the grocery store. Saves you not only time but also money”.

Parents mentioned that they appreciated the support, encouragement, and information from other parents who were in a similar life stage as they were, with seven (30%) commenting that they appreciated the support. One parent said, “It was helpful to see other parents offer their suggestions, so that I could try them as well”. Another parent said, “I would recommend a group like this, just for the support aspect alone. It’s good to hear how other moms do things with no judgement”. Table 3 includes additional responses from the open-ended responses.

## 4. Discussion

This study explored the feasibility of using Facebook to engage low-income parents and to provide nutrition education related to healthy eating on a budget. The results suggest that a majority of parents remained engaged throughout the 3-week intervention, as indicated by a 90% interaction rate with the last post. Participant retention over the study period was also high. Acceptability data indicated that the Facebook-adapted version of the Sesame Street curriculum was well-received by low-income parents. Parents gained knowledge from the content and from fellow peers participating in the Facebook group, as indicated by the responses to the open-ended questions. The majority of participants indicated that they would recommend joining a similar Facebook group to other parents in a similar life-stage.

Data from the current study show that Facebook can connect parents, particularly those living in rural areas, by fostering interactivity among users. Parents expressed that they appreciated the support they received from their peers and that this was a highlight of the Facebook group. They shared similar experiences, tips, and suggestions with each other related to eating healthy on a budget. Head Start parents are interested in being more involved and receiving more social support; however, they are often unable to overcome common barriers, such as time and transportation constraints [25]. Social support is important for the well-being of low-income parents because it increases overall parental functioning and psychological well-being [26]. Social media provides a means of support for parents who may otherwise have limited opportunities to discuss their child’s eating behaviors, and it provides opportunities for parents to identify helpful feeding strategies for their young children [27]. In the current study, the support among parents was fostered by the provision of evidence-based resources and the presence of a moderator to facilitate conversations within the group.

An analysis of open-ended responses indicated that parents were interested in learning about eating healthy on a budget. Parents reported that they learned a lot from participation in the Facebook group, and they found posts about picky eating as one the most useful types of information presented. Although definitions and measures of picky eating vary, between 14–50% of parents identify their preschool-aged child as a picky eater [28,29,30]. Picky eating can also increase mealtime stress and impact meal preparation [31]; therefore, it is not surprising that picky eating tips were found to be the most helpful in the current study. Together, these data suggest that it is advantageous to teach parents, especially parents who perceive their child to be a “picky eater”, that healthy food like fruits and vegetables can be low-cost, convenient, and liked by children [32]. 

Despite the Facebook group being well-liked by participants, there were individual differences in the rates of participant engagement. This may be due to individual differences in Facebook familiarity and typical frequency of use [33]. In addition, we did not assess the device used to access the Facebook group, and differences may have been seen between participants using a mobile device versus a computer at a library due to differing levels of accessibility. In the current study, a majority of participants reported logging into Facebook daily, while 32 percent reported that they visited the Facebook group multiple times per day. Because our intervention dosage was limited to 2–3 posts/day, participants who only logged on once may have missed multiple posts. We did not assess familiarity with Facebook prior to the study, so it is plausible that some participants were not familiar with using and navigating the platform. Future studies should investigate why some parents were more engaged than others to better inform how to increase parent engagement in social media behavioral interventions.

The results suggested that participant engagement, as defined by the number of interactions, varied by post type and content. Posts including discussion questions had the highest number of comments, and posts including videos received the highest number of likes. Similarly, Swindle et al. found that views and interactions with posts varied based on the type and content of posts, with posts containing links having the highest number of interactions [34]. Another study that examined whether different types of posts differentially affected participant engagement in a behavioral weight loss intervention found that posts that contained polls and encouraged votes resulted in the most participant engagement [35]. The current study and others did not report the time participants spent viewing the videos and posts; therefore, it is unknown whether participants truly engaged with the content as intended. More advanced web analytics should be used to gain more insight into the level of participant engagement. Future studies should examine whether post type engagement varies by content type, which would inform which types of posts are most impactful before designing interventions to be delivered on Facebook. 

Posts initiated by participants received a high number of likes and comments, suggesting that parents want to hear and learn from each other. In addition, participants mentioned that they appreciated hearing from other parents and that they liked the support they received from others who were in a similar life stage. Peer led discussions may be beneficial for prompting participant engagement and promoting learning. A study examining the peer-based Grow2Gether Facebook group (a social media parenting group) found that peers tended to provide information that was sound and helpful when mothers posed direct questions regarding infant health and that mothers were eager to both ask and answer questions in the Facebook setting [36]. Although peer groups may have utility for delivering intervention comments, one challenge is the need for constant monitoring [37]. In the current study, one participant was removed for posting inappropriate content, so a system for monitoring appears to be necessary. Our monitoring protocol worked as intended and was feasible to use with this study. However, the protocol was labor intensive, so automated monitoring of social media groups should be explored as an alternative. Future research should further examine the utility of peer led groups for delivering intervention content. 

This study had a number of limitations. First, the sample was small and homogenous, and the findings therefore may not be generalizable to diverse populations. Study participants were part of a convenience sample, which may have led to selection bias. Another limitation was the study length. Due to the relatively short length, little is known regarding whether or not engagement would be impacted with a longer intervention period. Additionally, the findings are specific to Facebook and may not be generalizable to other social media platforms. Finally, due to the limitations of using Sociograph with a closed group, some of the features were not available to us and limited our understanding of the data. However, for the research purposes of this group, a closed group was required to protect the identities of the participants. 

## 5. Conclusions

In summary, the data from this study indicate that social media platforms such as Facebook provide a feasible mode for delivering nutrition education and engaging low-income families living in rural areas. Facebook shows promise as a tool to engage parents, and it allows messages to be tailored based on the needs of the community members participating in the program. As new interventions are developed to reach low-income populations, social media should be explored as a platform to provide education and conduct interventions with this population.

## Figures and Tables

**Figure 1 ijerph-19-01457-f001:**
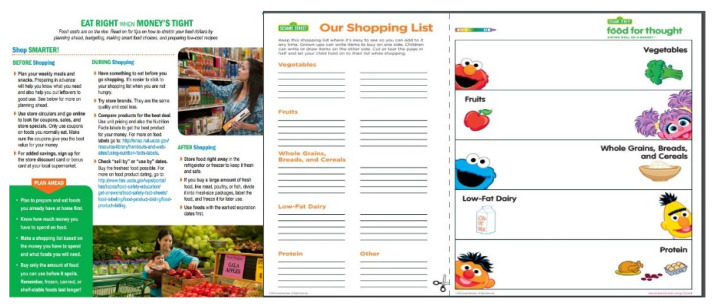
Materials provided to participants in the Facebook group.

**Table 1 ijerph-19-01457-t001:** Participant Characteristics.

Characteristic	n (%)
Employment Status	
Full time	4 (16 %)
Part-time	3 (12%)
Stay-at-home parent	12 (48%)
Unemployed	2 (8%)
Did not answer	4 (16%)
Annual Household Income	
<$10,000	4 (16%)
$10,000–$19,999	3 (12%)
$20,000–$29,999	8 (32%)
$30,000–$39,999	4 (16%)
$40,000–$49,999	1 (4%)
Did not answer	5 (20%)
SNAP Benefits	20 (80%)
WIC	12 (48%)
TANF	5 (20%)
Relationship to the Child	
Parent	22 (88%)
Foster parent	1 (4%)
Did not answer	2 (8 %)
Caregiver’s Race	
White, Non-Hispanic	21 (84%)
White, Hispanic or Latino	1 (4%)
Did not answer	3 (12%)
Education	
Some high school	1 (4%)
High school graduate	9 (36%)
Some college/technical school	12 (48%)
Completed college	1 (4%)
Did not answer	2 (8%)

**Table 2 ijerph-19-01457-t002:** Engagement by Intervention Post-Type.

Post Type	Number of Posts	Likes (Mean)	Comments (Mean)	Seen by (Mean)
Informative	3	9	3	24
Link (Handout or Printable Materials)	5	8.6	1	22.8
Video	6	10.5	2.7	24.5
Poll	3	2	3.3	25
Question	9	4.9	9.9	25.8

**Table 3 ijerph-19-01457-t003:** Select Participant Responses to Open-Ended Questions.

Question 1. Would you recommend joining a similar group to other caregivers of Head Start Preschoolers? Why or why not?
“I would recommend a group like this, just for the support aspect alone. It’s good to hear how other moms do things with no judgement”. “I think that parents can greatly benefit from being connected to other parents with similar issues”. “If I heard someone struggling with a picky eater, or where to find a farmers market, I would recommend Food for thoughts. Lots of info on there, and peers to answer questions”. “This group is friendly and it has amazing info for the parents that struggle to get their kids to try new food”. “It helped me to better budget, plan, spend more time with my kids, and got my kids eating better. I think every parent should be a part of something like this if they are not already”.
Question 2. What was the most useful thing you learned?
“That it really does make a difference to meal plan for the upcoming week and stick to that plan when at the grocery store. Saves you not only time but also money”. “I learned a few tips from the other parents on how to get my picky eater to try new foods”. “About the different farmers markets in different areas as well as other parent’s ideas on getting kids to try new foods”. “How to better budget while meal planning and making multiple meals from one meal”. “Ways to involve the kids in meal planning”.
Question 3. Are you doing anything new with your family that you were not doing before you joined the secret Facebook group? Please explain.
“… I plan to start making a ‘menu’ or meal plan for the week or maybe two and buy what is needed. I do plan to go back and look at the recipes the group has provided and try at least one but hopefully more”. “Trying new recipes and new ways to try to get them to eat new foods”. “Trying different healthier foods and recipes for my family. Trying to limit certain foods and instill healthier eating for my family”. “I am trying to get all my children to eat anytime foods rather than sometime foods”. “Kids are helping more with cooking and preparing meals”.

## Data Availability

Not applicable.

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
