# Peer review of "Feasibility of Using Facebook to Engage SNAP-Ed Eligible Parents and Provide Education on Eating Well on a Budget"

_ijerph, 2022, doi:10.3390/ijerph19031457_

Round 1

Reviewer 1 Report

Dear Authors:

The manuscript with the title “Feasibility of Using Facebook to Engage SNAP-Ed Eligible Parents and Provide Education on Eating Well on a Budget” focus a very pertinent subject. The Introduction is well written with the subject well contextualized. Methods are clearly described and Results clearly presented. On Discussion, authors gave special attention to the main limitations of this study. I have only some minor comments to address:

Comment 1:

On Material and Methods, it is important that authors organize it in sub-chapters. Thus authors must organize Material and Methods the following way:

2.1. Participants (from lines 70 to 84)

2.2. Evaluations (from lines 85 to 112).

2.2. Intervention (from lines 113 to 139)

2.3. Data analysis (from lines 140 to 169)

Comment 2:

Line 244: Authors must change “Qualitative” by “Responses to the open-ended questions”.

Author Response

Comment 1:

On Material and Methods, it is important that authors organize it in sub-chapters. Thus authors must organize Material and Methods the following way:

2.1. Participants (from lines 70 to 84)

2.2. Evaluations (from lines 85 to 112).

2.2. Intervention (from lines 113 to 139)

2.3. Data analysis (from lines 140 to 169)

Response: Thank you for the suggestion. We reorganized the "Materials and Methods" section into sub-sections (lines 70-177)

Comment 2:

Line 244: Authors must change “Qualitative” by “Responses to the open-ended questions”.

Response: Changed "Qualitative" to "Responses to open-ended questions" (line 251)

Reviewer 2 Report

Introduction:

Well written with a good structure. A little long winded given the lack of direct literature in the area but the section does well to highlight the need investigation the current intervention.

Line 42: “children ages 3-5.” Add 3-5 years.

Method:

A little hard to navigate in its current form – consider the use of relevant subtitles (participants, protocol, treatment of data etc.) to allow the reader to better pinpoint specifics details.

Lines 94-96 seem to be a direct repetition of lines 91-92 (consider revising / combining).

Include the relevant reference to article / analysis for claim on Line 121-123.

This section could benefit from relevant supporting images if available.

Analysis approach (thematic analysis) is very basic and not specific to what this research team specifically did to analyse the data. Could there be more in sight to the tailored approach of the 6 stages (plus constant comparison method) used towards this data set?

Results:

A bit basic but the reporting the descriptive statistics is sufficient given the complementary qualitative approach. That said, quantitive “engagement” with the posts doesn’t necessarily mean that the participants are employing the information that they are given. I think this needs to be appreciated more within the manuscript and this needs to lead to some more cautious interpretation about what the data show and the subsequent conclusions made.

Discussion:

As with my above comment, the authors do briefly note upon the fact that the length of video views would have provided better insight. This is true, but this only scratches the surfaces of the limited conclusions we can draw. If participants were asked to post evidence of meals / shopping within the group this would provide better insight to not only engagement but which elements were most useful (etc.).

Authors could note upon the fact that some individuals use Facebook / social media via apps on mobile devices while others use it on laptops.  The former would naturally lead to more individuals being more likely to engage more often (given accessibility), while allowing them to potentially take such information with them to supermarkets more readily to carry out the suggestions from the intervention. This might be worth noting?

All in all, this section needs to have a more meaningful discussion about what has been learned in this feasibility study and what (methodologically) needs to taken forward for future research.

Author Response

Thank you for your suggestions. Please see the attachment with our responses.
